# Virulence Traits and Azole Resistance in Korean *Candida auris* Isolates

**DOI:** 10.3390/jof9100979

**Published:** 2023-09-28

**Authors:** Seung A. Byun, Yong Jun Kwon, Ga Yeong Lee, Min Ji Choi, Seok Hoon Jeong, Dokyun Kim, Min Hyuk Choi, Seung-Jung Kee, Soo Hyun Kim, Myung Geun Shin, Eun Jeong Won, Jong Hee Shin

**Affiliations:** 1Department of Laboratory Medicine, Chonnam National University Medical School, Gwangju 61469, Republic of Korea; tmddk777@naver.com (S.A.B.); garei09@gmail.com (Y.J.K.); mkrkdud@naver.com (G.Y.L.); sjkee@jnu.ac.kr (S.-J.K.); alpinboy@chonnam.ac.kr (S.H.K.); mgshin@chonnam.ac.kr (M.G.S.); 2Microbiological Analysis Team, Biometrology Group, Korea Research Institute of Standards and Science (KRISS), Daejeon 34113, Republic of Korea; minji1246@naver.com; 3Department of Laboratory Medicine, Research Institute of Bacterial Resistance, Yonsei University College of Medicine, Seoul 03722, Republic of Korea; kscpjsh@yuhs.ac (S.H.J.); kyunsky@yuhs.ac (D.K.); tcmhwd@yuhs.ac (M.H.C.); 4Department of Laboratory Medicine, Asan Medical Center, University of Ulsan College of Medicine, Seoul 05505, Republic of Korea

**Keywords:** *Candida auris*, virulence, azole resistance

## Abstract

We analyzed the virulence traits and azole resistance mechanisms of 104 *Candida auris* isolates collected from 13 Korean hospitals from 1996 to 2022. Of these 104 isolates, 96 (5 blood and 91 ear isolates) belonged to clade II, and 8 (6 blood and 2 other isolates) belonged to clade I. Fluconazole resistance (minimum inhibitory concentration ≥32 mg/L) was observed in 68.8% of clade II and 25.0% of clade I isolates. All 104 isolates were susceptible to amphotericin B and three echinocandins. In 2022, six clade I isolates indicated the first nosocomial *C. auris* cluster in Korea. Clade II *C. auris* isolates exhibited reduced thermotolerance at 42 °C, with diminished in vitro competitive growth and lower virulence in the *Galleria mellonella* model compared to non-clade II isolates. Of the 66 fluconazole-resistant clade II isolates, several amino acid substitutions were identified: Erg11p in 14 (21.2%), Tac1Ap in 2 (3.0%), Tac1Bp in 62 (93.9%), and Tac1Bp F214S in 33 (50.0%). Although there were a limited number of non-clade II isolates studied, our results suggest that clade II *C. auris* isolates from Korean hospitals might display lower virulence traits than non-clade II isolates, and their primary fluconazole resistance mechanism is linked to Tac1Bp mutations.

## 1. Introduction

*Candida auris* is an emerging multidrug-resistant yeast closely related to *Candida haemulonii* within the family Metschnikowiaceae [1,2]. *C. auris* was initially reported in 2009 from the ear cultures of one Japanese patient and 15 Korean patients and was subsequently isolated from the blood cultures of three patients in Korea [1,3,4]. Since then, *C. auris* has been identified in over 40 countries across six continents as a causative agent of healthcare-associated infections, presenting significant public health concerns due to nosocomial outbreaks and notable patient mortality [5,6,7]. Whole-genome sequencing studies have delineated five phylogenetically distinct clades of *C. auris*, each differing from the others by >200,000 single-nucleotide polymorphisms: clade I (South Asia), clade II (East Asia), clade III (South Africa), clade IV (South America), and clade V (Iran) [8,9,10,11,12]. Although these isolates from each continent are clonal, those from different continents form genetically distinct clades [8]. Multilocus sequence typing (MLST)-based phylogenetic analysis also demonstrated the differentiation of clades I–IV among global isolates [11,13,14,15]. The vast geographical spread and genetic diversity among *C. auris* isolates from different clades, which vary in virulence, pathogenicity, resistance level, resistance mutation, and genome, underscore the importance of extensive research across all clades [11,16,17,18,19].

Our prior study demonstrated that all 61 *C. auris* isolates from Korean hospitals were identically classified under the MLST cluster ST2, aligning with the East Asian clade (clade II) [11]. Clade II Korean isolates differ from other clades due to their comparatively lower antifungal resistance rates and fewer *ERG11* mutations linked with fluconazole resistance [11]. Interestingly, clear nosocomial clusters of fungemia have yet to be observed in Korea, with the majority (93%) of Korean *C. auris* isolates being derived from ear specimens [11]. The reasons that clade II *C. auris* isolates in Korean hospitals are not linked with hospital fungemia outbreaks remain unclear. Although local fungal epidemiology and infection control systems may provide partial explanations, it is plausible that clade II inherently differs from other clades in its potential for nosocomial transmission [20]. 

Hotspot mutations in the drug target gene *ERG11* have been identified as the primary mechanism for fluconazole resistance in *C. auris*, with some clade specificity [6,8]. In our prior study, 62.3% of a total of 61 isolates were found to be fluconazole-resistant (FR), but only 3 isolates carried a known azole-conferring mutation in *ERG11* (K143R) [11]. This finding suggests that other azole resistance mechanisms may be active in *C. auris* clade II. Several recent studies have demonstrated that gain-of-function mutations in the transcription factor gene *TAC1*, which regulates the expression of drug efflux pumps, are contributors to azole resistance in *C. auris*; other mechanisms may also contribute [21,22,23,24,25,26]. Despite the continued isolation of FR *C. auris* from Korean hospitals, genetic determinants in target or transcription factor genes in Korean isolates of *C. auris* have yet to be completely identified.

In this study, we explored the potential virulence traits of clade II *C. auris* isolates, including thermotolerance, biofilm formation, in vitro competitive growth, and virulence in the *Galleria mellonella* model. Then, we compared these traits with those of other clade isolates of *C. auris* and closely related species such as the *C. haemulonii* complex and *Candida pseudohaemulonii*. Finally, we investigated the antifungal susceptibility profiles and amino acid substitutions (AASs) of *ERG11*, *TAC1A*, and *TAC1B* in these isolates to evaluate their relationships with azole resistance. To our knowledge, this study is the first to examine differences in virulence traits and azole resistance mechanisms between clade II isolates of *C. auris* and isolates of other clades by examining a comprehensive collection of clinical *C. auris* samples from Korean hospitals.

## 2. Materials and Methods

### 2.1. Fungal Isolates and Antifungal Susceptibility Testing

We collected 104 non-duplicated clinical isolates of *C. auris*, which consisted of 96 clade II (5 blood and 91 ear) and 8 clade I (6 blood and 2 other) isolates, from 104 patients in 13 Korean hospitals from 1996 to 2022. We extended the sample collection period to 1996 because our Korean collection contained the first bloodstream isolate recovered from a Korean hospital in 1996. This isolate was incidentally found by the molecular identification of unidentified yeasts, which were recovered in 1996 as bloodstream isolates [3]. No *C. auris* isolates were detected in the 1996–2005 period, but *C. auris* isolates have been recovered from ear cultures continually from several hospitals in Korea since 2006 [11]. Additionally, 20 species closely related to *C. auris*, including 11 *C. pseudohaemulonii*, 7 *Candida haemulonii* sensu stricto, and 2 *Candida haemulonii* var. *vulnera*, were sourced from Korean hospitals during the same period. All isolates were identified by sequencing the internal transcribed spacer (ITS) region and/or the D1/D2 regions of 26S ribosomal DNA within their rRNA genes [3,27]. In vitro antifungal susceptibility tests for fluconazole, amphotericin B (AMB), anidulafungin, caspofungin, micafungin, voriconazole, itraconazole, posaconazole, and flucytosine were conducted using the Sensititre YeastOne system (SYO; Thermo Fisher Scientific, Waltham, MA, USA) [28]. Two reference strains (*Candida parapsilosis* ATCC 22019 and *Candida krusei* ATCC 6258) were used as quality control isolates in each test. Minimum inhibitory concentration (MIC) endpoints in the SYO system were visually assessed after 24 h of incubation based on color changes (purple or blue) in comparison to drug-free growth (red). Categorical results were derived based on tentative MIC breakpoints for *C. auris* provided by the US Centers for Disease Control and Prevention (CDC): AMB, ≥2 mg/L; fluconazole, ≥32 mg/L; anidulafungin, ≥4 mg/L caspofungin, ≥2 mg/L; and micafungin, ≥4 mg/L (https://www.cdc.gov/fungal/candida-auris/c-auris-antifungal.html (accessed on 10 Sep 2023)). For comparative purposes, we also tested a panel of 10 *C. auris* isolates from each of the four clades, sourced from the CDC and Food and Drug Administration (FDA) Antimicrobial Resistance Isolate Bank. The CDC-FDA isolates included 5 isolates of clade I (AR0382, AR0387, AR0388, AR0389, and AR0390) and 1 isolate of clade II (AR0381), 2 isolates of clade III (AR0383 and AR0384), and 2 isolates of clade IV (AR0385 and AR0386) [11]. The Institutional Review Board of Chonnam National University Hospital approved this study (IRB CNUH-2020-117). Due to the retrospective and in vitro nature of this study, the requirement for informed consent was waived. 

### 2.2. MLST

MLST was conducted on all 104 isolates of *C. auris*, as previously described (Appendix A) [11,29]. The four genes selected for MLST analyses included *RPB1*, *RPB2*, and those encoding the ITS and D1/D2 regions. Amplified products underwent purification using a PCR purification kit (GeneAll Biotechnology, Seoul, Republic of Korea). The purified reagents were analyzed using an ABI PRISM 3730XL analyzer (Applied Biosystems, Waltham, MA, USA) with the same primers employed for the PCR. Two *C. auris* strains from clade II (KCTC 17809 and KCTC 17810) were used as controls [11].

### 2.3. Thermotolerance Assay

Thermotolerance was assessed by plating yeasts (1–10 × 10^11^ colony-forming units [CFU]) on Sabouraud dextrose agar (SDA) using the quadrant-streaking modified method, based on a previously described method [30]. Growth was evaluated after 3 days of incubation on SDA at temperatures of 30 °C, 37 °C, 40 °C, 42 °C, and 45 °C. Following incubation, growth was recorded as having reached the first (+), second, third (++), or fourth quadrant (+++). The in vitro growth rates of *C. auris* isolates at 42 °C were also determined using European Committee on Antimicrobial Susceptibility Testing (EUCAST) microplates, adhering to the specified inoculum and medium conditions [31]. The growth curves for each isolate (1 × 10^6^ CFU/mL) were plotted by incubating the EUCAST plates at 42 °C with moderate shaking in a spectrophotometer (Thermo Scientific, Waltham, MA, USA). For each well, the optical density at a wavelength of 490 nm (OD_490_) was recorded hourly for up to 24 h.

### 2.4. Biofilm Formation and Aggregate-Forming Capacity

Biofilm formation was examined using a denture strip model combined with an XTT reduction assay, as described previously [32,33]. Briefly, 200 μL of yeast suspension (5 × 10^7^ cells) in phosphate-buffered saline (PBS) was dispensed into 12-well cell culture plates (Corning Costar, Corning, NY, USA) and mixed with 1.8 mL PBS. A 1.5 cm^2^ denture strip (diameter, 15 mm; Nunc 174969; Thermanox Coverslips, Naperville, IL, USA) was placed in each well containing the yeast–PBS mixture. The cells were allowed 90 min for adherence at 37 °C. After adherence, non-adherent cells were gently washed off the strips with PBS, and the strips were transferred to new 12-well polystyrene microtiter plates. These strips were submerged in 4 mL of a 1× yeast nitrogen base medium (Becton Dickinson, Franklin Lakes, NJ, USA) supplemented with glucose and incubated at 37 °C for 24 h. Parallel controls were also prepared using only the medium. After incubation, these strips were transferred to a new 12-well plate containing 4 mL of PBS mixed with 50 μL of XTT [2,3-bis(2-methoxy-4-nitro-5-sulfophenyl)-5((phenyl amino) carbonyl)-2H-tetrazolium hydroxide; Sigma-Aldrich, St. Louis, MO, USA] and 4 μL of 1 mM menadione (Sigma-Aldrich). The plates were shielded with aluminum foil and incubated at 37 °C for 5 h. The amount of XTT formazan in the supernatant was quantified at 492 nm using a spectrophotometer. The aggregate-forming capacity was assessed for all 104 isolates, as previously described method [17]. A single *C. auris* colony was emulsified in approximately 20 μL of sterile water on a microscope slide and was examined microscopically at ×400 magnification. 

### 2.5. In Vivo Virulence Analysis Using Galleria Mellonella

Virulence was assessed by observing the survival rate of *G. mellonella* after infection with the isolates at 24, 48, 72, and 96 h post-infection at 37 °C, based on a previously described method with slight modifications [34]. Briefly, 20 sixth-instar *G. mellonella* larvae (S-worm, Cheonan, Republic of Korea) weighing 150–200 mg were selected for each test group. Each larva was infected with 10 μL of *C. auris* (1 × 10^5^ cells/larva) using a 50 μL Hamilton syringe fitted with a 25-gauge needle via inoculation through the hindmost right proleg. Following infection, the larvae were placed in sterile Petri dishes and incubated at 37 °C. The survival rates of the larvae were recorded every 24 h over a 4-day period. Three control groups were set up, in which larvae were injected with 10 μL of phosphate-buffered saline (PBS) (*n* = 20), subjected to needle puncture only (*n* = 20), or remained untouched (*n* = 20). After confirming that all control groups were alive at the end of experiments, larval mortality was determined if they exhibited no response to touch and had a darkened cuticle. This procedure was replicated three times, with 20 larvae in each group. Data from all trials were combined to compute the average percentage of survival.

### 2.6. In Vitro Competitive Growth Assay

We developed a method for the in vitro assessment of competitive growth in *C. auris* based on a previously described approach with modifications [35]. We examined a total of 84 *C. auris* isolates, including 75 Korean isolates (63 ear, 11 blood, and 1 urine isolate) and 9 CDC strains not belonging to clade II. Briefly, the strains were cultivated overnight in a Muller–Hinton broth at 30 °C. Each isolate was mixed in a 1:1 ratio with the *Candida albicans* SC5314 strain, and their quantities were equalized using a hemocytometer. The mixture, including the Muller–Hinton broth, was cultured for 24 h at 35 °C. Both *C. auris* and *C. albicans* were quantified in each mixture before (at inoculation) and after the 24 h incubation period. For CFU enumeration, the mixtures were diluted in Muller–Hinton broth (1:100), and 10 μL of the diluted mix was subcultured on ChromAgar *Candida* (Becton Dickinson, Heidelberg, Germany). After 48 h of incubation at 35 °C, the colonies of *C. auris* (pink or dark purple) and *C. albicans* (green) were identified visually. The competition growth index was calculated as follows: Competition index = (24 h incubated *C. auris* CFU/24 h incubated *C. albicans* CFU)/(Inoculated *C. auris* CFU/Inoculated *C. albicans* CFU) [36]. Each isolate underwent six individual tests.

### 2.7. Sequence Analysis of ERG11, TAC1A, and TAC1B

The sequence analyses of *ERG11*, *TAC1A*, and *TAC1B* were performed as described previously [11,23,37]. Genomic DNA was isolated from each isolate using the DNeasy Plant Mini Kit (Qiagen, Hilden, Germany, Germantown, MD, USA) and employed as a template for PCR amplification. The primers used for both the amplification and sequencing of *ERG11*, *TAC1A*, and *TAC1B* have been previously described (Appendix A). PCR products underwent purification using a PCR purification kit (GeneAll Biotechnology, Seoul, Republic of Korea), following the manufacturer’s protocol. Both strands of the PCR products were sequenced using an ABI PRISM 3130XL genetic analyzer (Applied Biosystems, Waltham, MA, USA). Nucleotide sequence analysis was performed using the MegAlign software package (DNAStar, Inc., Lasergene, Madison, WI, USA). The sequences from each isolate were aligned and analyzed against the reference sequences *ERG11* (GenBank accession no. XM_018315289.1), *TAC1B1* (B9J08_004820), and *TAC1A* (B9J08_004819) [11,37,38].

### 2.8. Statistical Analyses

Quantitative data are represented as the means ± standard deviation (SD). Categorical data are expressed as counts and proportions. For categorical data, comparisons were made using the χ^2^ or Fisher’s exact test. Student’s t-test and the Mann–Whitney U test were used for parametric or non-parametric testing, respectively. The kinetic parameters of isolates are expressed as medians and interquartile ranges and were compared at each time point based on their inclusion/non-inclusion in clade II using the Wilcoxon matched-pairs signed rank test. Data processing and analyses were performed using GraphPad Prism v9.3.1 (GraphPad Software Inc., San Diego, CA, USA). Statistical significance was determined using a threshold of *p* < 0.05.

## 3. Results

### 3.1. MLST Analysis and Antifungal Susceptibility Testing

Of the 104 isolates of *C. auris* examined using MLST analysis, 96 (5 blood and 91 ear) were identified as belonging to clade II, whereas 8 (6 blood, 1 urine, and 1 ear) belonged to clade I. Table 1 shows the results of antifungal susceptibility tests for 104 *C. auris* isolates (96 from clade II and 8 from clade I), 11 *C. pseudohaemulonii*, and 9 *C. haemulonii* complex isolates (7 *C. haemulonii sensu stricto* and 2 *C. haemulonii* var. *vulnera*) from Korean hospitals, against AMB, fluconazole and three echinocandins: voriconazole, itraconazole, posaconazole, and flucytosine. Based on the tentative MIC breakpoints for *C. auris*, the resistance rates for fluconazole (MIC ≥ 32 mg/L) were 68.8% (66/96) for clade II *C. auris*, 25.0% (2/8) for clade I *C. auris*, and 44.4% (4/9) for *C. haemulonii* complex isolates. By contrast, the resistance rate was 0% (0/11) for *C. pseudohaemulonii* isolates.

All 96 clade II *C. auris* isolates registered AMB MICs of ≤1 mg/L (indicating AMB susceptibility), and 75% (6/8) of clade I *C. auris* had AMB MICs of 2 mg/L. The AMB resistance rates (MIC ≥ 2 mg/L) were 100% for *C. pseudohaemulonii* and 77.8% (7/9) for *C. haemulonii* complex isolates. All 124 isolates of *C. auris* and its closely related species were found to be susceptible to the three echinocandins. The geometric mean MICs for AMB and the three echinocandins were higher for clade I *C. auris* isolates than for clade II *C. auris* isolates (AMB: clade I vs. clade II 1.682 mg/L vs. 0.461 mg/L; anidulafungin: 0.352 mg/L vs. 0.084 mg/L; caspofungin: 0.271 mg/L vs. 0.066 mg/L; and micafungin: 0.145 mg/L vs. 0.050 mg/L; all *p* < 0.0001). The geometric mean MICs for voriconazole, itraconazole, and posaconazole were significantly higher in FR clade II *C. auris* isolates than in fluconazole-susceptible (FS) clade II *C. auris* isolates (voriconazole by 14.1-fold; itraconazole by 3.8-fold; posaconazole by 6.4-fold; all *p* < 0.0001). All *C. pseudohaemulonii* and *C. haemulonii* var. *vulnera* isolates had MICs of ≤ 0.12 mg/L for voriconazole, itraconazole, or posaconazole. Additionally, six clade II *C. auris* isolates (three FS and three FR) showed flucytosine MICs of 64 mg/L, whereas all isolates from other species had flucytosine MICs of ≤0.5 mg/L.

### 3.2. Thermotolerance and Biofilm Formation Assay

Table 2 presents the thermotolerance assay results for 124 isolates of *C. auris* and related species from Korean hospitals, as well as an additional 10 *C. auris* isolates from the CDC collection. The growth rate of all 97 clade II (96 from Korean hospitals and 1 from CDC collection [AR0381]) isolates of *C. auris* was similar at 30 °C. They proliferated at 30 °C and 37 °C on SDA plates, but their growth was diminished at 42 °C. In addition, 10 of the 95 clade II isolates grown at 40 °C and 67 of 85 isolates grown at 42°C did not reach the +++ level of the quadrant, respectively. Specifically, 11 (11.3%) isolates, all derived from ear cultures, failed to grow at 42 °C, and none of the 97 isolates grew at 45 °C on SDA plates. Conversely, all eight clade I *C. auris* isolates from Korean hospitals and all nine non-clade II isolates from the CDC collection proliferated at 30 °C, 37 °C, 40 °C, and 42 °C. Among these 17 non-clade II isolates, 14 (6 from Korea and 8 from the CDC) grew at 45 °C, although the extent of growth was weaker than the lower temperature. However, all 20 *C. pseudohaemulonii* and *C. haemulonii* isolates grew only at 37 °C, with none growing at 40 °C, 42 °C, or 45 °C.

A thermotolerance assay of all 114 *C. auris* isolates using EUCAST plates at 42 °C showed that the mean growth rate of the 97 clade II *C. auris* isolates was significantly lower than that of the 17 non-clade II isolates [average growth rates: 2.03 × 10^−7^ s^−1^ for clade II vs. 4.18 × 10^−7^ s^−1^ for non-clade II] (Figure 1). The minimum required OD value of 0.2 was reached within approximately 11 h for non-clade II, compared to an average of 15 h for clade II isolates. The peak growth rates of the isolates were observed at 11 h for non-clade II and 18 h for clade II. The growth peak of clade II isolates was lower than those of non-clade II isolates (mean ± SD: 0.596 ± 0.293 for clade II vs. 0.866 ± 0.338 for non-clade II; *p* < 0.0001). This pronounced difference in the average growth between the two clades was evident between 8 h and 21 h.

The average biofilm formation ability of the 96 clade II *C. auris* isolates from Korean hospitals, as determined using the XTT reduction assay, was similar to that of the 8 clade I *C. auris* isolates and the 9 non-clade II isolates from the CDC collection (Figure 2A). Similarly, the average biofilm formation ability of the 104 *C. auris* isolates was comparable to that of the 11 *C. pseudohaemulonii* isolates but four-fold that of *C. haemulonii* complex isolates (7 *C. haemulonii sensu stricto* and 2 *C. haemulonii* var. *vulnera*) (mean ± SD: 0.407 ± 0.1497 for *C. auris* vs. 0.106 ± 0.095 for *C. haemulonii* complex; *p* < 0.0001) (Figure 2B). In addition, almost all clade I and clade II isolates of *C. auris* from Korean hospitals were non-aggressive, and only three ear (two FR and one FS) isolates of clade II were aggregative (2 FR and 1 FS isolates).

### 3.3. In Vivo Virulence Analysis Using G. mellonella and In Vitro Competitive Growth Assay

The average 96 h survival rates of *G. mellonella* infected with clade II *C. auris* isolates were significantly greater than those infected with clade I *C. auris* isolates from Korean hospitals (27.6% for clade II vs. 9.1% for clade I; *p* < 0.0001). Notably, all isolates of *C. pseudohaemulonii* and *C. haemulonii* exhibited higher 96 h survival rates compared to all *C. auris* isolates (75.3% and 89.3%, respectively vs. 18.3%; *p* < 0.001) (Figure 3A).

To assess the competitive growth between clade I and II clinical isolates of *C. auris* from Korean hospitals, each isolate was subjected to competition experiments against the *C. albicans* SC5314 strain. The outcomes of these competition experiments are summarized in Figure 3B. The mean competition index of the 67 clade II *C. auris* isolates from Korean hospitals, obtained after 24 h of co-culture with the *C. albicans* SC5314 strain, was significantly lower (mean ± SD: 0.1904 ± 0.1085) than that of 17 non-clade II *C. auris* isolates (mean ± SD: 4.274 ± 2.065; *p* < 0.0001, Mann–Whitney U test). Within the clade II group, no significant difference in the competitive index was observed based on fluconazole resistance (mean ± SD: 0.1842 ± 0.1031 for FR isolates of clade II vs. 0.2089 ± 0.1244 for FS isolates of clade II; *p* = 0.4008).

### 3.4. Sequence Analysis of ERG11, TAC1A, and TAC1B

Table 3 shows the results of the antifungal susceptibility testing and sequencing of *ERG11*, *TAC1A*, and *TAC1B* genes for 18 *C. auris* isolates, including 8 clade I isolates of *C. auris* and 10 CDC isolates. Among the two FR clade I isolates, one (#1) had Erg11p K143R and Tac1Bp A640V, matching two clade I CDC isolates (AR0388 and AR0390), whereas the FR clade I ear isolate (#2) showed W188C in Tac1Ap and G91C in Tac1Bp. Interestingly, all six clade I isolates (5 blood and 1 urine) from a 2022 collection (#3–8) were FS and lacked Erg11p, Tac1Ap, and Tac1Bp AASs.

Table 4 presents the sequencing results for *ERG11*, *TAC1A*, and *TAC1B* genes in 96 clade II isolates of *C. auris* from Korean hospitals. Although two Tac1Ap AASs (V145E and K713N) and one Tac1Bp AAS (K215R) were identified in all 96 clade II *C. auris* isolates, they are not listed. Among the 96 clade II isolates, 17.7% (16/96) had 9 Erg11p AASs, 2.1% (2/96) had 2 Tac1Ap AASs, and 72.9% (70/96) had 26 Tac1Bp AASs. The most frequent Erg11p AAS was K143R (6 FR and 1 FS isolates, 7.3%), and Tac1Bp AAS F214S was the most prevalent (33 FR and 1 FS isolates, 35.4%), followed by L582I (6 FR and 1 FS isolates, 7.3%). Among the 66 FR isolates, 14 (21.2%) had 8 Erg11p AASs (L43H, Y132F, K143R, R157S, A298G, Q357K, I466L, and E468V), 2 (3.0%) had 2 Tac1Ap AASs (I783F and E281Q), and 62 (93.9%) had 21 Tac1Bp AASs (R50L, C168F, F191V, S192N, W198C, W198L, S201Y, R202M, F214S, N393I, S427R, R505S, Q526E, S531P, L582I, P595A, P595L, F649I, A651V, P860T, and D865N). Notably, three Tac1Bp AASs (S192N, R202M, and P860T) were identified in both FS and FR isolates. Twelve of these FR isolates had a combination of Erg11p, Tac1Ap, or Tac1Bp AASs. Five FR isolates with fluconazole MIC ≥ 64 mg/L had both Tac1Bp F214S and either Erg11p (with K143R, R157S, or A298G mutations) or Tac1A (with I783F mutation) AASs.

## 4. Discussion

Five clades of *C. auris* have independently emerged in various regions globally, but both local and international transmissions have been reported in many countries, with human movement and international travel as major transmission vectors [38,39]. In this study, we conducted an MLST analysis on 104 isolates collected from Korean hospitals from 1996 to 2022. Of these, 96 were identified as clade II and 8 as clade I. Clade I *C. auris* isolates were characterized in 6 bloods, 1 ear, and 1 urine sample from three separate hospitals. Our previous study identified only clade II *C. auris* isolates [11]. Thus, our new findings indicate the emergence of clade I *C. auris* isolates in Korean hospitals starting in 2018. Intriguingly, six clade I isolates (5 blood and 1 urine) collected from a single hospital in 2022 exhibited the same or very similar antifungal susceptibility patterns. To our knowledge, this study is the first to report a hospital-based cluster caused by clade I *C. auris* isolates in Korea.

Surveillance has indicated that approximately 90% of *C. auris* isolates are resistant to fluconazole, 50% to AMB, and 5% to echinocandins, according to tentative CDC breakpoints [7,8,16,40]. Furthermore, multidrug resistance spanning two or more antifungal classes has been observed in >30% of clinical samples [7,8,16,40]. In our study, none of the 96 clade II isolates from Korean hospitals demonstrated resistance to AMB or the three echinocandins, and fluconazole resistance was observed in 68.8% of isolates. These deviations in resistance patterns could stem from the specific clade II distribution within Korean hospitals, echoing the findings of our prior study [11]. A significant proportion of *C. auris* isolates in clade I, which is often termed the South Asian clade, that was sampled from India and Pakistan displayed resistance to fluconazole (97–100%) with varied resistance to polyenes (7.9–93.7%) [41,42,43]. However, most of our clade I isolates (75%) from Korean hospitals were FS and registered an AMB MIC of 2 mg/L. Given the reports of FS clade I *C. auris* isolates in Brazil, China, and Taiwan [21,44,45], our data suggest the international transmission of FS clade I isolates in these regions, including Korea. Notably, geometric mean MICs for AMB and the three echinocandins tested with the SYO system were significantly higher in clade I *C. auris* isolates from Korean hospitals than in clade II isolates. This finding suggests that clade I may have inherently higher AMB and echinocandin MICs than clade II, although further study with a larger sample size could provide clearer insights.

Phylogenetically, *C. auris* is related to the *C. haemulonii* complex and *C. pseudohaemulonii* [27,46,47,48]. The *C. haemulonii* complex includes *C. haemulonii* sensu stricto, *C. duobushaemulonii*, *C. vulturna*, and *C. haemulonii var. vulnera*. Although *C. pseudohaemulonii* is phylogenetically close, it is not currently classified within the *C. haemulonii* complex [27,46,47]. These species frequently exhibit resistance to azoles and AMB [21,46]. However, a recent study indicated that a majority (44/46) of *C. haemulonii* complex strains from the US and Brazil were susceptible to azoles [49]. Additionally, all four *C. pseudohaemulonii* from the US were susceptible to both AMB and azoles, whereas two of the four isolates were resistant to the three echinocandins [49]. In our study, AMB and fluconazole resistance were observed in 77.8% and 44.4% of *C. haemulonii* complex samples from Korean hospitals, respectively. All 11 *C. pseudohaemulonii* isolates were resistant to AMB but susceptible to azoles and echinocandins. Collectively, these findings imply that both the *C. haemulonii* complex and *C. pseudohaemulonii* could rapidly develop resistance to AMB, azoles, or echinocandins, leading to regional variation in antifungal susceptibility among these isolates.

Thermotolerance in *C. auris* has been linked to environmental adaptation in response to rising global temperatures, leading to a global warming emergence hypothesis [50,51]. In this study, the *C. haemulonii* complex and *C. pseudohaemulonii* isolates from Korean hospitals did not grow at 40 °C, whereas almost all *C. auris* isolates thrived at this temperature, which is consistent with the findings of other studies [30,50]. Clearly, *C. auris* flourishes at 42 °C, making thermal tolerance a distinguishing feature of *C. auris* compared with other *Candida* species [6,50]. Thus, this study is the first to reveal a variation in thermotolerance between clade I and II *C. auris* isolates at 42 °C, based on an analysis of a significant number of clade II isolates. All 17 non-clade II isolates tested in this study thrived at 42 °C, whereas 11.5% (11 ear isolates) of the clade II isolates from Korean hospitals did not grow at 42 °C on SDA plates. The average growth rate and maximum peak of 97 clade II *C. auris* isolates on EUCAST plates at 42 °C were significantly lower than those of the 17 non-clade II isolates. Given that nearly all clade II *C. auris* isolates from Korean hospitals were derived from the ear [5,11,52], which is cooler than the core body temperature, the superior thermotolerance of non-clade II isolates might enable them to endure on skin regions such as the axilla and groin, which are common *C. auris* isolation sites. This adaptation facilitates colonization and aids in the prolonged environmental persistence of non-clade II isolates of *C. auris* [6,53]. It has been previously known that *C. auris* presents two growth phenotypes, one aggregative and the other non-aggregative, presenting differences in virulence, resistance to disinfectants, and biofilm formation [17,34]. In this study, we found that almost all clade I and clade II isolates of *C. auris* from Korean hospitals were non-aggressive, and only three clade II (two FR and one FS) isolates from ear cultures were aggregative. These data supported the unique aspects of clade II *C. auris* isolates from Korean hospitals regarding phenotypical characteristics.

Survival and growth under competition with other commensal *Candida* species, which are part of the normal human skin and gut microbiota, could be essential for the microbial invasion and pathogenicity of *C. auris*. Considering that *C. albicans* is the most prevalent commensal *Candida* species responsible for invasive candidiasis, we developed a method to assess the competitive growth ability of each *C. auris* isolate. This method involved a 24 h co-culture in a 1:1 ratio with the *C. albicans* SC5314 strain at 35 °C. For the first time, here we showed that the mean competitive growth ability of clade II *C. auris* isolates from Korean hospitals was 5.2-fold lower than that of the *C. albicans* SC5314 strain. In contrast, the growth ability of 17 non-clade II *C. auris* isolates was 4.3-fold higher than the *C. albicans* SC5314 strain. Moreover, clade II *C. auris* isolates from Korean hospitals were less virulent than other clade *C. auris* isolates in the *G. mellonella* model, though no significant differences in biofilm formation were observed between the two clade isolates. Although isolates from all five clades have been linked to invasive infections, clade II *C. auris* isolates from Korean hospitals stand out due to their lack of nosocomial transmission [11,52]. Therefore, we hypothesize that their reduced competitive growth ability compared to *C. albicans*, including diminished virulence in the *G. mellonella* model and lower thermal tolerance, might be factors associated with their limited colonization in hospital environments, multiple body sites, and absence of nosocomial transmission.

Several antifungal resistance mechanisms of *C. auris* have been addressed, such as mutations in hot spots of the target gene *ERG11*, the overexpression of efflux pumps (*CDR1*, *CDR2*, and *MDR1*), a gain of function mutations in the transcription factor, gene duplication, modifications in lipid content, biofilm formation or the progression of aging, respectively [21,22,23,24,25,26]. Here, we examined AASs in *ERG11*, *TAC1A*, and *TAC1B* of *C. auris* isolates from Korean hospitals to understand their associations with azole resistance. In this study, 13.6% and 51.5% of all FR isolates carried Erg11p (6 K143R, 1 Y132F, 1 L43H, and 1 Q357K) and Tac1Bp (33 F214S, and 1 P595L) AASs that were previously described in FR isolates [23,37]. Mutations in *C. auris* Erg11p K143R were significant contributors to fluconazole and voriconazole resistance, although the MIC of other triazoles showed minimal alterations [53]. In this study, the geometric mean MICs for voriconazole, itraconazole, and posaconazole were considerably higher against FR isolates of clade II *C. auris* compared to FS isolates of clade II *C. auris*, indicating the possible role of overexpression in multidrug efflux pump genes. Although the direct role of the newly identified AASs in conferring FR remains uncertain, the Erg11p, Tac1Ap, and Tac1Bp AAS that were novel in FR isolates were found in 7.6%, 3.0%, and 40.9% of the isolates, respectively. Overall, Erg11p, Tac1Ap, and Tac1Bp AASs were present in 21.2%, 3.0%, and 93.9% of the 66 FR isolates, respectively. Taken together, these findings imply that Tac1Bp mutations may be the predominant fluconazole resistance mechanism in clade II Korean isolates. 

The link between point mutations of *TAC1A* and *TAC1B* genes and the expression of drug efflux pumps remains an open question [37]. Several recent studies have shown that gain-of-function mutations in transcription factor genes of *TAC1* control the expression of drug efflux pumps, while some studies have indicated that *CDR1* expression is not notably affected by *TAC1A/TAC1B* deletion or the presence of the *TAC1B* mutation, suggesting the existence of Tac1-dependent and Cdr1-independent azole resistance mechanisms [23,24,37]. Our results show that in 66 FR isolates, 8 Erg11p AASs, 2 Tac1Ap AASs, and 21 Tac1Bp AASs were found either singly or in combination, without distinct hot-spot regions. Given that other mechanisms unexplored in this study are also possible, it is difficult to determine whether a certain *TAC1A/TAC1B* AAS is a GOF mutation without data from gene editing experiments. However, we found that Tac1Bp F214S was the most prevalent (35.4%) in our isolates, which has been associated with the elevated expression of the ABC-type efflux pump-encoding gene *CDR1* [23]. Rybak et al. demonstrated the role of the Tac1Bp F214S mutation in three fluconazole-evolved derivative strains and showed a corresponding increase in *CDR1* expression in relation to the parental clinical isolate [23]. We found that Tac1Bp F214S was detected in 34 (33 FR and 1 FS) isolates and a significant range of fluconazole MIC values for *C. auris* isolates possessed Tac1Bp F214S (8 to >256 mg/L). Moreover, five FR isolates with fluconazole MIC ≥ 64 mg/L contained both Tac1Bp F214S and another Erg11p (K143R, R157S, A298G) or Tac1Ap (I178F) AAS. Collectively, our observation supports the hypothesis that the combined presence of both *ERG11* and *TAC1B* mutations could have a cumulative effect, resulting in elevated fluconazole MIC values.

## 5. Conclusions

In this study, we examined the potential virulence characteristics and azole resistance mechanisms of clade II *C. auris* isolates from Korean hospitals based on an analysis of 104 patient isolates collected from 13 Korean hospitals during 1996–2022. Their virulence traits revealed that clade II isolates of *C. auris* from Korean hospitals exhibit reduced thermal tolerance at 42 °C, reduced virulence in the *G. mellonella* model, and a reduced competitive growth ability compared to non-clade II isolates. These factors may at least partially contribute to the decreased colonization by clade II *C. auris* in hospital settings and at various body sites and to their absence in nosocomial transmissions. Fluconazole resistance was observed in 68.8% of clade II isolates, with Tac1Bp mutations being the predominant mechanism of this resistance. Notably, this study identified six clade I isolates of *C. auris* from a single hospital in 2022, indicating the emergence of the first nosocomial cluster due to *C. auris* in Korea. Given the potential for further introductions and the continued nosocomial spread of non-clade II *C. auris* isolates, there is a pressing need to enhance laboratory capabilities to distinguish clade I and non-clade II *C. auris* isolates, and improved infection control measures in Korean hospitals are also required.

## Figures and Tables

**Figure 1 jof-09-00979-f001:**
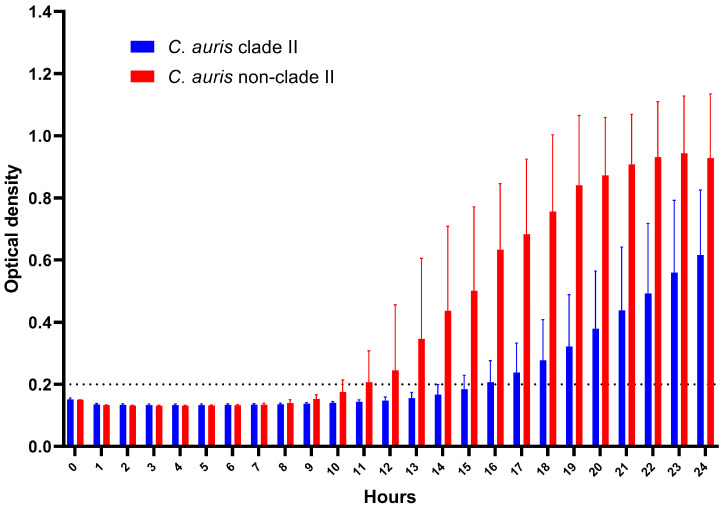
Kinetic growth curves at 42 °C for 97 clade II isolates (96 Korean and 1 CDC isolate) (blue bars) and 17 non-clade II isolates (8 clade I Korean and 9 non-clade II CDC isolates) (red bars) of *C. auris*. These data were obtained following the European Committee on Antimicrobial Susceptibility Testing method, but with an incubation temperature of 42 °C.

**Figure 2 jof-09-00979-f002:**
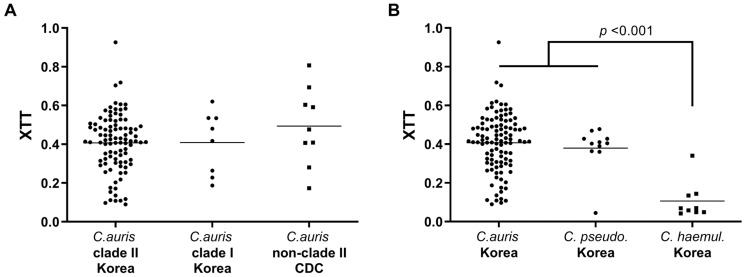
(**A**) Biofilm formation in *C. auris* clinical isolates (clades I and II) from Korea in addition to non-clade II CDC-FDA isolates. (**B**) Biofilm formation in *C. auris* clinical isolates from Korean hospitals compared to those of closely related species, assessed using an XTT assay. Abbreviations: *C. pseudo.*, *C. pseudohaemulonii*; *C. haemul.*, *C. haemulonii* complex.

**Figure 3 jof-09-00979-f003:**
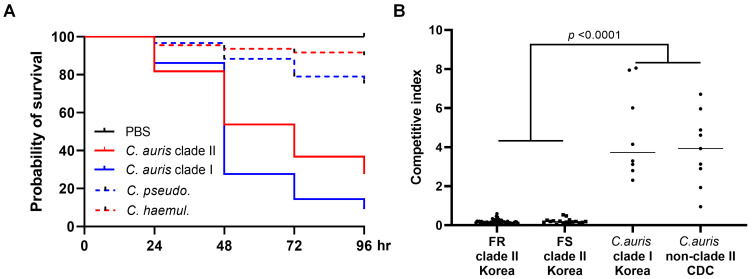
(**A**) In vivo virulence of *C. auris* clinical isolates (clades I and II) compared to those of closely related species, assessed using the *Galleria mellonella* infection model. (**B**) Competitive growth index of *C. auris* isolates obtained after 24 h of co-culture in a 1:1 ratio with the *Candida albicans* SC5314 strain at 35 °C. Abbreviations: PBS, phosphate-buffered saline; *C. pseudo.*, *C. pseudohaemulonii*; *C. haemul.*, *C. haemulonii* complex.

**Table 1 jof-09-00979-t001:** In vitro antifungal susceptibility patterns of *Candida auris* and closely related species collected from Korean hospitals.

**Agent**	**Species/Clade ***	**No. Tested**	**No. of Isolates with the Following MIC (mg/L) ****												**No. (%) of Resistant Isolates ^†^**	**Geometric Mean of MICs**
			≤0.03	0.06	0.12	0.25	0.5	1	2	4	8	16	32	≥64		
Antifungals which tentative MIC breakpoints were provided by the US CDC																
FLU	*C. auris* clade II	96						1	9	8	5	7	**25**	**41**	66 (68.8)	27.697
	*C. auris* clade I	8						3	1	1	1			**2**	2 (25.0)	6.727
	*C. pseudohaemulonii*	11						1	9	1					0 (0)	2.000
	*C. haemulonii*	7							2	1				**4**	4 (57.1)	40.317
	*C. haemulonii* var. *vulnera*	2							1			1			0 (0)	5.657
AMB	*C. auris* clade II	96			4	13	69	10							0 (0)	0.461
	*C. auris* clade I	8						2	**6**						6 (75)	1.682
	*C. pseudohaemulonii*	11							**4**	**1**	**6**				11 (100)	4.537
	*C. haemulonii*	7						1	**1**	**1**	**4**				6 (85.7)	5.040
	*C. haemulonii* var. *vulnera*	2						1	**1**						1 (50.0)	1.414
ANI	*C. auris* clade II	96	17	21	50	4	4								0 (0)	0.086
	*C. auris* clade I	8			1	4	1	2							0 (0)	0.352
	*C. pseudohaemulonii*	11		2	3	6									0 (0)	0.158
	*C. haemulonii*	7		1	4	2									0 (0)	0.153
	*C. haemulonii* var. *vulnera*	2	1		1										0 (0)	0.042
CAS	*C. auris* clade II	96	12	59	24	1									0 (0)	0.066
	*C. auris* clade I	8			1	5	2								0 (0)	0.271
	*C. pseudohaemulonii*	11		5	1	5									0 (0)	0.184
	*C. haemulonii*	7		3	3	1									0 (0)	0.144
	*C. haemulonii* var. *vulnera*	2	1			1									0 (0)	0.061
MCF	*C. auris* clade II	96	33	55	8										0 (0)	0.050
	*C. auris* clade I	8		1	4	3									0 (0)	0.145
	*C. pseudohaemulonii*	11	4	3	4										0 (0)	0.060
	*C. haemulonii*	7	3	3	1										0 (0)	0.071
	*C. haemulonii* var. *vulnera*	2	1	1											0 (0)	0.030
Antifungals which tentative MIC breakpoints were not provided by the US CDC																
VOR	*C. auris* clade II	96	20	7	6	26	25	6	5		1				NA	0.190
	FR (*n* = 66)			1	2	26	25	6	5		1				NA	0.435
	FS (*n* = 30)		20	6	4										NA	0.031
	*C. auris* clade I	8	5			1		1			1				NA	0.080
	*C. pseudohaemulonii*	11	10	1											NA	0.021
	*C. haemulonii*	7	3			1		1	1		1				NA	0.349
	*C. haem.*var. *vulnera*	2	1		1										NA	0.042
ITC	*C. auris* clade II	96	10	23	11	43	8	1							NA	0.140
	FR (*n* = 66)		2	6	7	42	8	1							NA	0.212
	FS (*n* = 30)		8	17	4	1									NA	0.056
	*C. auris* clade I	8	4		1	1	1		1						NA	0.112
	*C. pseudohaemulonii*	11	9	2											NA	0.030
	*C. haemulonii*	7		2	2		1	1				1			NA	0.436
	*C. haem.*var. *vulnera*	2	1	1											NA	0.042
POS	*C. auris* clade II	96	34	12	32	15	3								NA	0.066
	FR (*n* = 66)		6	10	32	15	3								NA	0.118
	FS (*n* = 30)		28	2											NA	0.018
	*C. auris* clade I	8	5		2			1							NA	0.047
	*C. pseudohaemulonii*	11	6	5											NA	0.021
	*C. haemulonii*	7	2	2		1	1				1				NA	0.164
	*C. haem.*var. *vulnera*	2	1	1											NA	0.021
5-FC	*C. auris* clade II	96	23	17	40	10								6	NA	0.139
	FR (*n* = 66)		18	14	26	5								3	NA	0.114
	FS (*n* = 30)		4	4	14	5								3	NA	0.211
	*C. auris* clade I	8	1	5	1		1								NA	0.085
	*C. pseudohaemulonii*	11		11											NA	0.060
	*C. haemulonii*	7		5	1	1									NA	0.081
	*C. haem.*var. *vulnera*	2		2											NA	0.060

* Each strain was characterized by multilocus sequence typing (MLST) with an ST cluster using an allele profile of four genes (ITS-*RPB1*-*RPB2*-D1/D2). MLST clusters ST1 and ST2 correspond to clade I and clade II, respectively [11]. ** Antifungal minimum inhibitory concentrations (MICs) were determined using the Sensititre YeastOne system (Thermo Fisher Scientific, Waltham, MA, USA). ^†^ The numbers (%) of isolates were categorized using tentative MIC breakpoints for *C. auris*, according to the US Centers for Disease Control and Prevention (CDC): FLU ≥ 32 mg/L, AMB ≥ 2 mg/L, ANI ≥ 4 mg/L, CAS ≥ 2 mg/L, and MCF ≥ 4 mg/L [5]. Numbers in bold indicate resistant isolates. Abbreviations: FLU, fluconazole; AMB, amphotericin B; ANI, anidulafungin; CAS, caspofungin; MCF, micafungin; VOR, voriconazole; ITC, itraconazole; POS, posaconazole; 5-FC, flucytosine; MIC, minimum inhibitory concentration.; NA, not applicable.

**Table 2 jof-09-00979-t002:** Thermotolerance of *Candida auris* and closely related species recovered from Korean hospitals and the CDC collection.

Source	Species/Clade	No. Tested	No. of Isolates Grown (No. with Growth Degree +++/++/+) at: *				
			30 °C	37 °C	40 °C	42 °C	45 °C
Korean collection	*C. auris*, clade II	96	96 (96/0/0)	96 (96/0/0)	95 (85/2/8)	85 (18/40/27)	0 (0/0/0)
	*C. auris*, clade I	8	8 (8/0/0)	8 (8/0/0)	8 (8/0/0)	8 (6/0/2)	6 (0/4/2)
	*C. pseudohaemulonii*	11	11 (11/0/0)	11 (11/0/0)	0 (0/0/0)	0 (0/0/0)	0 (0/0/0)
	*C. haemulonii*	7	7 (7/0/0)	7 (7/0/0)	0 (0/0/0)	0 (0/0/0)	0 (0/0/0)
	*C. haemulonii* var. *vulnera*	2	2 (2/0/0)	2 (2/0/0)	0 (0/0/0)	0 (0/0/0)	0 (0/0/0)
FDA- CDC collection	*C. auris*, clade II	1	1 (1/0/0)	1 (1/0/0)	1 (1/0/0)	1 (0/0/1)	0 (0/0/0)
	*C. auris*, clade I	5	5 (5/0/0)	5 (5/0/0)	5 (5/0/0)	5 (3/2/0)	4 (0/1/3)
	*C. auris*, clade III	2	2 (2/0/0)	2 (2/0/0)	2 (2/0/0)	2 (2/0/0)	2 (0/0/2)
	*C. auris*, clade IV	2	2 (2/0/0)	2 (2/0/0)	2 (2/0/0)	2 (2/0/0)	2 (0/2/0)

* Thermotolerance was evaluated by plating yeasts (1–10 × 10^11^ colony-forming units [CFU]) on Sabouraud dextrose agar (SDA) plates using the quadrant streaking method. Growth was recorded as having reached the first (+), second, third (++), or fourth quadrant (+++). Abbreviations: FDA-CDC, Food and Drug Administration and U.S. Centers for Disease Control and Prevention.

**Table 3 jof-09-00979-t003:** Results of antifungal susceptibility testing and *ERG11*, *TAC1A*, and *TAC1B* sequencing for 8 clade I isolates of *C. auris* from Korean hospitals and 10 isolates from the CDC collection.

*C. auris* Isolates	Clade	FLU-AMB-ANI-CAS-MIC Susceptibilities	Amino Acid Substitutions of *		
			Erg11p	Tac1Ap	Tac1Bp
**Korean collection (source, year, hospital)**					
1 (Blood, 2018, E)	I	R-S-S-S-S	**K143R**	None	**A640V**
2 (Ear, 2019, F)	I	R-S-S-S-S	None	W188C	G91C
3 (Blood, 2022, D)	I	S-S-S-S-S	None	None	None
4 (Blood, 2022, D)	I	S-S-S-S-S	None	None	None
5 (Blood, 2022, D)	I	S-S-S-S-S	None	None	None
6 (Blood, 2022, D)	I	S-S-S-S-S	None	None	None
7 (Blood, 2022, D)	I	S-S-S-S-S	None	None	None
8 (Urine, 2022, D)	I	S-S-S-S-S	None	None	None
FDA-CDC collection (AR No)					
9 (AR0381)	II	S-S-S-S-S	None	V145E, K713N	K215R
10 (AR0382)	I	S-S-S-S-S	None	None	None
11 (AR0383)	III	R-S-S-S-S	**F126L**	V13I, S116A, V145E, G149D, D500E, E560D, E565D, S627G, E758G, S762P, A766T	S36L, S89Y, E200K, K215R, K225N, Q226R, I268V, D278V, Q298K, C331S, C334F, L335S, S339A, T346I, Q503R, F580L, Y608H, Y647C, S754N, L774M, M809I, 772-773^*dle*^
12 (AR0384)	III	R-S-S-S-S	**F126L**	Identical to AR0383	Identical to AR0383
13 (AR0385)	IV	R-S-S-S-S	**Y132F**	V13I, S116A, V145E, A288S, D500E, E560D, S762P	S36L, K215R, Q226R, D278V, L328Q, C331S, C334F, L335S, S339A, Y608H, S754N, M809I, 862-866^*del*^
14 (AR0386)	IV	R-S-S-S-S	**Y132F**	Identical to AR0385	Identical to AR0385
15 (AR0387)	I	S-S-S-S-S	None	None	None
16 (AR0388)	I	R-S-S-S-S	**K143R**	None	**A640V**
17 (AR0389)	I	R-R-S-S-S	**Y132F**	None	L582I
18 (AR0390)	I	R-S-S-S-S	**K143R**	None	**A640V**

* Isolate sequences were compared and analyzed based on reference sequences for *ERG11* (GenBank accession no. XM_018315289.1), *TAC1B1* (B9J08_004820), and *TAC1A* (B9J08_004819) [11,23]. AASs previously detected that FR *C. auris* isolates are indicated in bold [16,37]. Abbreviations: FDA-CDC Food and Drug Administration and U.S. Centers for Disease Control and Prevention; AR No.; Antimicrobial Resistance Bank numbers; FLU, fluconazole; AMB, amphotericin B; ANI, anidulafungin; CAS, caspofungin; MCF, micafungin.

**Table 4 jof-09-00979-t004:** Sequencing results for the *ERG11*, *TAC1A*, and *TAC1B* genes in 96 clade II isolates of *C. auris* from Korean hospitals.

Amino Acid Substitutions of *			No. of Isolates with Fluconazole MIC (mg/L):					Total No.
Erg11p	Tac1Ap	Tac1Bp	≥64 (FR)	32 (FR)	16 (FS)	8 (FS)	≤4 (FS)	
None	None	**F214S**	16	11		1		28
None	None	S201Y, L582I	1	3				4
**K143R**	None	None	2		1			3
**K143R**	None	S427R	2	1				3
None	None	L582I	1	1				2
R157S	None	**F214S**	2					2
None	None	Q526E	2					2
None	None	W198C	1	1				2
**K143R**	None	**F214S**	1					1
None	I783F	**F214S**	1					1
A298G	None	**F214S**	1					1
None	None	N393I, D865N	1					1
**Q357K**	None	L582I	1					1
I466L	None	S192N, S427R	1					1
None	None	R50L	1					1
None	None	R202M, **P595L**	1					1
None	None	S531P	1					1
E468V	None	S531P	1					1
None	None	P595A	1					1
None	None	A651V	1					1
**Y132F**	None	None	1					1
None	None	W198L, R505S		3				3
**L43H**	None	**F214S**		1				1
None	None	C168F		1				1
None	None	F191V		1				1
None	None	F649I		1				1
None	E281Q	P860T		1				1
None	None	R202M			1			1
None	None	P600S			1			1
None	None	L704V			1			1
None	None	P860T				1		1
R462K	None	None					1	1
None	None	P213L					1	1
None	None	L759S					1	1
None	None	S192N, D845Y					1	1
None	None	None	1		3	3	14	21
Total			41	25	7	5	18	96

* Isolate sequences were compared and analyzed based on reference sequences for *ERG11* (GenBank accession no. XM_018315289.1), *TAC1B1* (B9J08_004820), and *TAC1A* (B9J08_004819) [11,23]. Amino acid substitutions (AASs) previously found in FR *C. auris* isolates are indicated in bold [16,37]. Three common Tac1Ap AASs (V145E and K713N), and 1 Tac1Bp AAS (K215R) were not listed because all clade II *C. auris* isolates had these AASs. A total of 29 new AASs, found in FS and F-SDD isolates of *C. auris* in this study, were deposited in GenBank (accession nos.: Erg11p, OQ744543-OQ744646; Tac1A, OR455355-OR455358; Tac1B, OR455359-OR455383). Among these new Tac1Bp AASs, P595A and A651V occurred at positions previously described in FR isolates [23]. Abbreviations: MIC, minimum inhibitory concentration; FR, fluconazole-resistant; FS, fluconazole-susceptible.

## Data Availability

All data generated or analyzed in this study are included in this published article, and the datasets are available from the corresponding author within the limits imposed by ethical and legal dispositions.

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
