# Peer review of "Virulence Traits and Azole Resistance in Korean Candida auris Isolates"

_jof, 2023, doi:10.3390/jof9100979_

Round 1

Reviewer 1 Report

The article by Byun et al. describes possible molecular mechanisms of azole resistance in multidrug resistant Candida auris clinical isolates in Korea. The article is an interesting read and provides insight into molecular mechanisms of azole resistance through point mutations in drug resistance genes ERG11, TAC1A, and TAC1B. The article is well written. Here are my comments:

1) The introduction will greatly benefit if the authors introduced what these drug resistance gene regulates. This will greatly benefit the readers. The authors described these genes in later part of the manuscript (the discussion section).

2)Authors stated that TAC1A and TAC1B regulates efflux pumps.Do point mutations in these genes enhance the expression of the efflux pumps in these isolates? Please clarify.

3)The authors should state that the molecular mechanisms described in this manuscript is not the only mechanism of resistance. Other mechanisms can also be possible. Previous studies have described azole resistance due to gene duplication in TAC1, CDR1, and ERG11. Further, aging in C. auris has also been described as a possible mechanism. These are described in the following studies:

https://www.ncbi.nlm.nih.gov/pmc/articles/PMC7927865/

https://pubmed.ncbi.nlm.nih.gov/30911079/

4)What primers were used for sequencing and MLST. For better clarity, can these be described in a table rather than just referencing them. Please clarify.

5)Were the growth rates of all strains similar at 30C? Please clarify.

Reviewer 2 Report

In this article, the authors collected C. auris isolates from 13 Korean hospitals between 1996 and 2022. The isolates belong mainly to Clade II, with a few Clade I isolates. In order to be able to compare Clade II isolates to the rest of the clades, they obtained some isolates from other clades from the Centers for Disease Control and Prevention and Food and Drug Administration. Authors analyzed different virulence traits as well as antifungal resistance and the mutations behind the resistance. However, not all the different clades were tested in all the experiments; therefore, authors should be cautious when extrapolating the observed results to all non-Clade II isolates. It is a well written article that increases our knowledge on the emergent pathogen C. auris, which will probably be a major challenge in the future in the nosocomial setting. Nevertheless, the authors should address some problems. It has been described that C. auris presents two growth phenotypes, one aggregative and other non-aggregative, presenting differences in virulence, resistance to disinfectants, biofilm formation, etc. It would be interesting to know if the isolated obtained by the authors present these two different forms and if these phenotypes could be correlated to differences observed in their assays. In addition, as C. auris was first described in 2009, authors should explain why they have decided to extend the sample collecting period to 1996, and how many C. auris isolates were actually recover from 1996-2009 period. Finally, some minor issues should be addressed before the manuscript is ready for its publication, as mentioned below:

Abstract, line 25 and 26: Taking into account the limited number of non-Clade II isolates studied-specially clade III and IV-, I will suggest the authors to reconsider the statement and don’t apply it to all non-Clade II isolates.

Line 99: please change “clade 2” to “clade II”

Materials and Methods: Why was the 1996-2022 period selected? C. auris was described first in 2009, how many of the isolated were identified as C. auris before 2009?

2.1. Section: Please state clearly in this section how many isolates from Korean hospitals belong to each clade, and how many isolates from each clade have been obtained form CDC-FDA.

2.1 Section, lines 87-89: It is mentioned in the text that “For comparative purposes, we also tested a panel of 10 C. auris isolates from each of the four clades, sourced from the CDC & Food and Drug 88 Administration (FDA) Antimicrobial Resistance Isolate Bank (nos. AR0381–AR0390) [11].” However, antifungal susceptibility data of these isolates was not included in the corresponding section of results, and only fluconazole resistant is mentioned in Table 4 for one isolate of clade II and 5 isolates of clade I of these samples.

2.4. Biofilm formation assay: Why did the authors used a denture strip for biofilm formation? This is not the regular biofilm formation assay used for yeast.

2.6. In vitro competitive growth assay: What was used as growth control? Do both Candida species grow at the same rate in the medium and condition tested? Could the differences seen in this assay between C. auris and C. albicans be due to different growth rates under the studied conditions and not due to a competition between them?

Table 1 and 2 could be fused into one table, as they represent the same assay performed with different antifungals.

Table 3. Authors used the quadrant streaking method to asses growth at different temperatures and recorded growth as +, ++ or +++ depending on the quadrant reached. However, the results on table 3 are given as growth/no growth, with no mention to the quadrants that the yeasts reached. Were no differences in the reached quadrants among the isolates for the same temperature? What was the aim of using the quadrant streaking method if differences in growth at the same temperature are not analyzed?

Figure 1. In order to allow direct comparison of the growth curve of isolates from Clade II against non-Clade II isolates, I will suggest converting the bars into a line with markers for each hour and having both groups in the same graph.

Figure 2. In the foot of the figure, it should be mentioned that graph A is also showing the biofilm formation capacity of CDC-FDA isolates.

Line 280: G. mellonella should not be written in italics, as the rest of the subheading already is.

Figure 3A. How do the control groups of G. mellonella look like? Did all the PBS injected control larvae survive? This data should be also included in the graph.

3.4. Section. Why was the data on ERG11, TAC1A, and TAC1B sequencing divided in two tables? What it is the rationale behind this? It is confusing, as some isolated are represented in both tables.

Lines 387 to 401: None of the yeast names are written in italics in this paragraph. Please correct this.

Line 466: Please indicate what CDR1 stands for.
